# The Simulated Characterization and Suitability of Semiconductor Detectors for Strontium 90 Assay in Groundwater

**DOI:** 10.3390/s21030984

**Published:** 2021-02-02

**Authors:** Graeme Turkington, Kelum A. A. Gamage, James Graham

**Affiliations:** 1Electronics & Electrical Engineering, James Watt School of Engineering, University of Glasgow, Glasgow G12 8QQ, UK; g.turkington.1@research.gla.ac.uk; 2National Nuclear Laboratory, Central Laboratory, Sellafield, Seascale, Cumbria CA20 1PG, UK; james.graham@nnl.co.uk

**Keywords:** strontium-90, geant4 simulation, radiation, detectors, cadmium-telluride, decommissioning

## Abstract

This paper examines the potential deployment of a 10 mm × 10 mm × 1 mm cadmium telluride detector for strontium-90 measurement in groundwater boreholes at nuclear decommissioning sites. Geant4 simulation was used to model the deployment of the detector in a borehole monitoring contaminated groundwater. It was found that the detector was sensitive to strontium-90, yttrium-90, caesium-137, and potassium-40 decay, some of the significant beta emitters found at Sellafield. However, the device showed no sensitivity to carbon-14 decay, due to the inability of the weak beta emission to penetrate both the groundwater and the detector shielding. The limit of detection for such a sensor when looking at solely strontium-90 decay was calculated as 323 BqL${}^{-1}$ after a 1-h measurement and 66 BqL${}^{-1}$ after a 24-h measurement. A gallium-arsenide (GaAs) sensor with twice the surface area, but 0.3% of the thickness was modelled for comparison. Using this sensor, sensitivity was increased, such that the limit of detection for strontium-90 was 91 BqL${}^{-1}$ after 1 h and 18 BqL${}^{-1}$ after 24 h. However, this sensor sacrifices the potential to identify the present radionuclides by their end-point energy. Additionally, the feasibility of using flexible detectors based on solar cell designs to maximise the surface area of detectors has been modelled.

## 1. Introduction

At decommissioning sites, leaks and spills of nuclear waste lead to the introduction of radionuclides into the groundwater table. The concentrations of radionuclides in the groundwater can vary depending on the inventory of radioactivity released and the impact of hydrological and chemical factors, such as advection, dispersion, and sorption. The concentrations of radionuclides across the site are currently monitored by sampling groundwater from boreholes [1]. The majority of beta emitters found at Sellafield are found in the separation area, where plutonium and uranium were recovered from spent Magnox fuel during reprocessing. In 2016, the highest annual average total beta found in a borehole was 105,553 BqL${}^{-1}$, with a maximum result of 124,000 BqL${}^{-1}$. These wells are particularly active due to their location next to legacy waste storage plants, from which a number of leaks occurred in the 1970s [1].

For decommissioning purposes, beta emitters are split into two categories, strong and weak, depending on their decay energy. In the strong category, strontium-90, ${}^{90}$Sr, is the single largest contributor to total beta activity and its activity maps with the total beta activity on the site. ${}^{90}$Sr is a fission product and decays via the emission of a beta particle which may have a maximum energy of 0.546 MeV. Its daughter, ${}^{90}$Y 90, itself decays by beta emission up to 2.28 MeV. The another strong beta emitter of interest is ${}^{137}$Cs. It is a fission product which decays by beta emission to the metastable isomer barium-137, which returns to its groundstate via the emission of a 0.661 MeV photon. At Sellafield, ${}^{137}$Cs is typically found in very low concentrations; only one sampled borehole’s annual average exceeded the World Health Organisation’s (WHO) drinking water guideline of 10 BqL${}^{-1}$ due to its propensity to adsorb strongly to the Sellafield subsurface. Naturally occurring ${}^{40}$K, with an end point energy of 1.311 MeV, may also be found on site. Weak beta emitters emit at much lower energies and are, therefore, more difficult to detect. This category includes tritium with end point energy of 0.019 MeV, ${}^{14}$C which has an end point energy of 0.156 MeV. ${}^{14}$C can be produced in nuclear reactors and is found in boreholes in the Sellafield separation area, with the highest annual average concentration being 43,825 BqL${}^{-1}$; however, only 10 wells exceeded the WHO drinking water limit [1].

This research investigates the potential real-world performance of a cadmium telluride, CdTe, detector which will be deployed directly into a groundwater borehole. Current methods for monitoring ${}^{90}$Sr contamination require manual sampling of water from boreholes and analysis in laboratories with techniques, such as ionic separation and liquid scintillation counting [2]. This can take weeks to produce results, is time consuming and produces secondary waste. The proposed detector will directly measure ${}^{90}$Sr contamination in the groundwater after its insertion into groundwater boreholes, thereby providing results in a more timely and cost effective manner. The detector in this research has been designed to be sensitive to ${}^{90}$Sr decay [3] and utilises a 10 mm × 10 mm × 1 mm CdTe sensor supplied by Acrorad (Uruma City, Japan) [4]. This detector has an Ohmic configuration and operates with a bias voltage of 80 V. CdTe is a semiconductor compound with established use as an X-ray detector and can be used in solar panels [5,6]. The atomic number of CdTe is 50, and it has a band gap of 1.44 eV. This means that it is effective at absorbing ionising radiation and has potential to work at room temperature [6].

Developments in solar cell technology have seen some solar cells developed which are thin and flexible [7,8,9], and it may be possible to leverage these as radiation detectors [10]. It is not necessary to apply a bias voltage to the cells, which can be useful to reduce power requirements, and some research has found use for solar cell dosimeters in highly radioactive environments, such as the Fukushima Daiichi nuclear power station [11,12]. In particular, thin film detectors may have applications in medical physics applications Flexible solar cells have value in the energy industry because they require fewer and cheaper materials to manufacture, and “roll to roll” production techniques are used to reduce production costs [13]. As the cells are flexible, they can easily be applied to form factors which differ from common flat panel configurations. The cells are created when thin layers of CdTe, Si, or other semiconductor materials are deposited on pliable substrates. In fact, CdTe solar cells have become rather cheap to produce, and their efficiency is competitive with silicon cells [9]. They are often covered with a flexible glass or polyimide layer which is not only robust but provides some protection from water. One of the difficulties with monitoring groundwater in-situ is developing the detector form factor to easily fit within 5–6 cm diameter borehole. These properties could be exploited to create a compact detector for borehole monitoring by flexing the sensor into a cylindrical shape to maximise the surface area of the detector.

This research examines which radionuclides in-situ detector designs are sensitive to whether their sensitivity is useful to decommissioning authorities. Beta particles have a short range in matter which correlates with their decay energy which has presented difficulties when it comes to designing beta detectors. To determine the effectiveness of an in-situ semiconductor detector, this research has simulated the decay of commonly found radionuclides in a borehole scenario to find the limits of detection for a semiconductor detector and the time frame required to achieve such results. The sensitivity of the detector to common beta emitting radionuclides has been evaluated to examine whether they will play a significant role in the response function of the detector to contaminated groundwater.

## 2. The Detection of Typical Beta Emitters Found at Nuclear Decommissioning Sites

This detector is being designed to specifically look for ${}^{90}$Sr contamination in groundwater; however, ${}^{90}$Sr is unlikely to be found in isolation. Realistically, other radionuclides will be present, and this may include ${}^{137}$Cs, ${}^{14}$C, ${}^{40}$K, and tritium. As outlined above, these radionuclides emit radiation at very different energies and may be distributed differently throughout the environment due to their sorption properties [14]. Therefore, it is important to consider the sensitivity of the detector to the radionuclides likely to be found in the groundwater at decommissioning sites. As it is difficult to identify the origin of a beta particle, it is important to know which radionuclides are contributing to the total counts seen in the detector.

### 2.1. Simulation Layout

Geant4 is a Monte Carlo simulation code developed to examine the interaction between radiation and matter [15]. Written in C++, Geant4 code has been used to simulate radiation-matter interactions in high-energy particle physics, as well as medical and nuclear applications. The software tracks radiation, step-by-step, as it travels through matter. At the beginning of a step, a number is randomly generated, and this is used to determine whether the particle interacts with matter, and in what way, or otherwise proceeds along its path. When particles interact with the sensitive layer of the detector, the information about that interaction is stored and can be used for analysis. The physics used in the simulation are described by “physics lists”, which are predefined mathematical models that are used to calculate the outcome of interactions. This simulation used the FTFP BERT (Fritiof String Bertini Cascade) model physics list, which allows electron interactions up to 100 TeV, uses a maximum step length of 0.65 mm, an energy threshold of 900 keV, and a low energy limit of 1 eV. The simulation procedure tracks each individual particle from the beginning of their path to the end, before moving on to the next, and, therefore, does not account for the simultaneous arrival of particles on the detector.

A simulation of the performance of the detector in a borehole environment was modelled with a Geant4 simulation, Figure 1. A section of a groundwater borehole has been coded to model the surrounding soil, plastic piping, groundwater, and the detector casing and sensor. A cylinder with a 5 cm diameter and 5 cm height was modelled and populated sufficient radionuclides to match likely values found at Sellafield. In this simulation, a 1-h counting time was simulated for each concentration of radionuclide. The decaying atoms are randomly populated throughout the groundwater. In this simulation, the detector consists of a 10 mm × 10 mm × 1 mm CdTe detector with 20-nm thick platinum Ohmic contacts on either side. The detector is sealed by two layers of low density polyethylene (LDPE), a thin and waterproof plastic which protects the sensor and electronics from the groundwater but also attenuates incoming beta particles. The LDPE layers are 13.6 μm thick and have a density of 0.94 g cm${}^{-3}$. There is a 1-mm gap between the sensor and the LDPE which is filled with air at atmospheric pressure. The sensor is positioned in contact with the water, where it counts beta particles which strike its surface and records the energy they deposit as they are either fully absorbed or scatter before this can happen. This reflects the reality of current prototypes, where buoyancy of the detector sees it floating on the water surface.

### 2.2. Sensitivity to Radionuclides

Some of the pure beta emitting radionuclides found at Sellafield are ${}^{90}$Sr, ${}^{90}$Y, ${}^{14}$C, and naturally occurring ${}^{40}$K. It is important to establish the detector sensitivity to these radionuclides to establish whether they contribute to the detector response function. This will give insight into the viability of this detector in the real world and give insight into the difficulty of examining ${}^{90}$Sr decay in the presence of other radionuclides. In this section, the detector has been exposed to each of these radionuclides in a scenario which represents an exposure time of 1 h. The number of counts in the detector for different radionuclides at different activities is displayed in Table 1.

The detector is unable to detect any ${}^{14}$C counts, even at high concentrations of 100,000 BqL${}^{-1}$. This indicates that maximum decay energy of the radionuclide is an important factor in how sensitive the detector is to the radionuclide. Betas emitted by ${}^{14}$C are of such low energy, on average 0.049 MeV, that they are absorbed in the groundwater or detector casing before they are able to reach the sensor. If the detector is blind to this activity it will have some significant implications. It means that a total beta count of high and low energy emitters is not possible. However, this could be advantageous as weak betas are automatically filtered out of the detector response function, meaning it is only sensitive to the higher energy emitters which are of primary interest. The data shows that there is sensitivity to ${}^{90}$Sr, ${}^{90}$Y, ${}^{137}$Cs, and ${}^{40}$K in this scenario. It can be observed that the higher the energy of the decay, the more sensitive the detector is to the radionuclide. However, ${}^{40}$K is only found in almost negligible quantities at Sellafield, with a maximum result of 1.0 BqL${}^{-1}$ in 2016 [1]. It is not certain whether ${}^{90}$Sr and ${}^{90}$Y would exist in secular equilibrium as data is not collected to monitor ${}^{90}$Y concentrations in groundwater. This could be a significant factor in detector performance as ${}^{90}$Y decay overlaps with ${}^{90}$Sr and ${}^{137}$Cs decay in energy. If these radionuclides are found in secular equilibrium, and their activity is 100,000 BqL${}^{-1}$, the number of ${}^{90}$Y counts observed is 20.28 times that of ${}^{90}$Sr. This is due to the higher average emission energy of ${}^{90}$Y and its likelier transmission through water. ${}^{137}$Cs is typically found at low activities in the separation area of Sellafield, in the region of 10–40 BqL${}^{-1}$, and, as such, is likely going to be undetected and only add to the noise in the detector. The characteristic peak from the 32 keV decay is visible in the collected spectrum, enabling the identification of the radionuclide when it is present in sufficient activities. In conclusion, it is expected that, when the detector is deployed into groundwater, it will not be sensitive to weak beta emitters, such as ${}^{14}$C, but has sensitivity to strong betas, like ${}^{90}$Sr, ${}^{90}$Y, ${}^{40}$K, and ${}^{137}$Cs.

## 3. Low Detectable Limit

The limit of detection of a detector (LOD) describes the fewest number of counts which can be detected reliably by the system. This property is particularly relevant in the case of monitoring radionuclide activity at decommissioning sites as it describes the lowest activity which can be detected in a defined time frame. There are two aspects to consider when defining the detection limits of a detector, whether the sample is radioactive or not, and whether we can quantitatively measure its activity. This section shall consider the limit of detection for the sensor and the duration measurement required to achieve it.

In a simple case where a decision must be made whether a borehole contains radioactivity, the number of counts from the sampled water, $N_{T}$, and the number of counts from the background, $N_{B}$, are subtracted to produce a result, $N_{S}$, which can be compared with a critical level, $L_{C}$. This is used to determine the likelihood of measured counts representing true activity and not fluctuations in statistical noise. The minimum detectable activity, α, is calculated with the following equation, where $N_{D}$ is defined as the minimum number of counts needed to ensure that the detector does not produce a false-negative rate that exceeds 5% (an industry standard value) [16].
(1)$$N_{D}=4.653\sigma_{N_{B}}+2.706\;\mathrm{and}\;\alpha =\frac{N_{D}}{feT},$$
where *f* is the radiation yield per disintegration, *e* is the absolute detection efficiency, and *T* is the time taken to count the sample. When this formula is applied to ${}^{90}$Sr and ${}^{90}$Y, the expected low detectable limits are 323 BqL${}^{-1}$ and 35 BqL${}^{-1}$, respectively. The WHO guideline value for ${}^{90}$Sr contamination in water is 10 BqL${}^{-1}$. This suggests that, in its current state, the detector would be unsuited for monitoring radiation close to or at the guideline value. Performance could be improved by eliminating some of the background noise, which can be achieved by refining the electronic design and accounting for temperature control systems in the detector. The background noise for the detectors was collected while they were at room temperature, but the real world application will see them deployed in groundwater, which typically has a temperature of 7 ${}^{\circ }$C. A reduction in background noise of 50% would reduce the lower detectable limit by 29%, to 229 BqL${}^{-1}$, leaving it still significantly short of the safe drinking water limit. Alternatively, the counting time for the detector could be increased. The results of this are documented in Table 2. By increasing the measurement time to 24 h, the LOD would be significantly reduced, to 66 BqL${}^{-1}$, while still obtaining results on a daily basis. The background noise in the detector would have to be significantly reduced, by up to 98%, for the detector performance to reach the safe water drinking limit. However, daily sampling is far from the norm in the industry, where monthly sampling is considered high frequency. Longer time frame measurements are, therefore, much more viable, and this would allow the detector to produce results closer to the guidance level.

In this scenario, there are some peculiarities observed in the spectra displayed in Figure 2. A peak is observed at low energies where an excess of low energy particles are recorded by the detector. This is in part explained by the attenuation of beta particles as they interact with matter, the water they are found in, and the casing of the detector. The source of this attenuation is the coulombic interactions which beta particles are subject to as they travel through matter. Additionally, some beta particles backscatter on the detector surface and only deposit a fraction of their energy. This combination of factors distorts the spectrum and skews it towards lower energy particles.

An ideal detector with 100% efficiency would collect beta spectra with identical characteristics, no matter the activity of the water. The maximum recorded energy in the detector decreases as the activity decreases. As the activity goes from 100,000 BqL${}^{-1}$ to 10,000 BqL${}^{-1}$, the maximum recorded energy moves from 0.514 MeV to 0.403 MeV, a decrease of 21.6%. This shift increases to 25.5% and 41.0% for 10,000 BqL${}^{-1}$ and 1000 BqL${}^{-1}$, respectively. A similar shift in ${}^{90}$Y spectra is seen, although to a less significant extent. In that case, the maximum energy shifts by 7% from the maximum at 100,000 BqL${}^{-1}$ to 22.8% at 1000 BqL${}^{-1}$. This is the result of the statistics at play. In a typical beta spectrum, few particles are emitted at the end point energy, and, in this scenario, fewer still are detected.

These results suggest that the detector is unlikely to compete with traditional methods for monitoring ${}^{90}$Sr contamination in terms of precision. If truly instant results are required, the detector could be applied to monitor activity levels in highly concentrated areas above 800 BqL${}^{-1}$. By increasing the counting time to 168 h, or more, the detector can assess lower activity levels which approach the guideline value for ${}^{90}$Sr contamination in water. This would still produce results in quicker fashion than traditional techniques and would allow for rapid scanning of large areas to identify possible spikes in activities or new leaks. Operating over such a time period may present additional challenges, such as detector stability over time [17], changes in temperature, and power requirements.

## 4. Gallium-Arsenide (GaAs) Detector

The previous sections assesses a 10 mm × 10 mm × 1 mm CdTe sensor, but other sensors types are available, and one such sensor is a gallium-arsenide, GaAs, sensor with a surface area of 4 cm${}^{2}$ and a much thinner sensitive layer at 3 μm. This sensor is designed for solar panel usage and, therefore, has a larger surface area. This detector may be applied for the purposes of radionuclide monitoring, and its larger surface area, different material, and thinner sensitive layer provide a contrast in approach to the CdTe sensor.

The same simulations were repeated with the GaAs sensor. The range of activities found at Sellafield for radionuclides are shown in Table 3, with the percentage increase compared to the counts in the CdTe sensor. Notably, the counting rate is much higher, up to 344%, and this is due to the larger surface area of the sensor. This is not directly proportional to the increase in detector surface area as the GaAs detector is much thinner, 3 μm, allowing some betas to pass through undetected. This becomes particularly pertinent at the activities from 1–100 BqL${}^{-1}$. The GaAs detector is proportionally more sensitive to ${}^{90}$Y than ${}^{90}$Sr decay. However, in spite of the larger surface area, the GaAs detector is still unable to detect ${}^{14}$C decay in this scenario and is sensitive to the same radionuclides, ${}^{90}$Sr, ${}^{90}$Y, ${}^{137}$Cs, and ${}^{40}$K. It should be noted that the detector is sensitive to ${}^{14}$C decay in principle, if not in practice.

If it is assumed that the same level of background is present for the GaAs sensor, the LOD is significantly lower when compared to the CdTe detector. Table 4 describes the LOD of the GaAs sensor as the measurement time is increased. This means it is a more viable detector for monitoring contamination close to the safe water drinking limit, and results can be achieved within a 48-h period. The detector is more sensitive when it comes to simply counting the beta decay in the groundwater; however, the response function of the detector to radiation is significantly different to the CdTe sensor. In existing groundwater sampling and analysis techniques, ${}^{90}$Y decay may be counted as a method of estimating ${}^{90}$Sr activity in groundwater [18]; however, the in-situ activity of ${}^{90}$Y in the groundwater is not recorded. This is due to the weeks it takes for ${}^{90}$Sr analysis to happen and the short decay time of ${}^{90}$Y. It is unclear whether ${}^{90}$Y and ${}^{90}$Sr occur in groundwater in secular equilibrium or whether this ratio is disrupted by other factors, such as their sorption onto different materials in the ground. Figure 3 plots the detector response to a 10:1 ratio ${}^{90}$Sr and ${}^{90}$Y and their combined spectra. As previously indicated, the detector is more sensitive to ${}^{90}$Y decay, and there are approximately twice as many ${}^{90}$Y counts as ${}^{90}$Sr counts in these spectra. Figure 4 plots the spectrum captured by the detector for a groundwater mixture containing 10,000 BqL${}^{-1}$ of ${}^{90}$Sr and 1000 BqL${}^{-1}$ of ${}^{90}$Y. There is little distinction between the two sources in terms of the detector response, and it would be impossible to identify the presence of both radionuclides by looking at the energy deposits. The detector would lose its ability to distinguish between ${}^{90}$Sr and ${}^{90}$Y decay. This can be seen by comparing Figure 3 with Figure 4. The spectrum produced in the GaAs detector is compressed and shifted to the left, and only a few counts of ${}^{90}$Y 90 exceed the energy deposit produced by ${}^{90}$Sr. This reduces the possibility of identifying radionuclides by their energy deposit in the detector. As a further point of comparison, another GaAs detector was modelled with dimensions of 35 m × 35 mm × 0.03 mm. This detector would be capable of measuring down to 18 BqL${}^{-1}$ in 10 min, the equivalent performance taking 24 h in a smaller and thinner detector.

## 5. Flexible Detectors

Earlier results have demonstrated the importance of maximising the surface area of the detector in order to acquire results with a lower minimum detectable limit, and this section looks at an alternative detector design which can be used to increase the surface area of the detector.

The concept in this simulation is to have a detector with a cylindrical shape, encapsulating the electronics inside, and the flexible sensor wrapped around the surface of the cylinder. Flexible silicon detectors have been previously used by researchers to develop an alpha detector which can be deployed in pipes at decommissioning sites [19]. As flexible solar cells have become a mature technology, it may be possible to appropriate them as radiation detectors. This section models the layers of a CdTe cell from Salavei et al. [8] as an example detector. The CdTe solar cells consist of a 50-nm Au back contact, a 6-μm CdTe layer, a 300-nm cadmium sulphate layer, a 400-nm indium tin oxide and 100-nm zinc oxide front contact, and a 100-μm layer of glass. The detectors were protected by the same layers of LDPE as mentioned in previous simulations.

The detectors were deployed in the same groundwater borehole geometry as described in previous sections. In this scenario, the volume of the detector would displace the equivalent volume of groundwater as it is submerged. The detector would, therefore, be exposed to a smaller volume of water which fills the gap between the outer layer of the detector and the plastic filter pack which makes up the groundwater borehole. This scenario is illustrated in Figure 5.

Four detector geometries were simulated. Detectors 1 and 3 had surface area dimensions of 30 mm × 15 mm and were wrapped around a cylinders with radii of 15 mm and 20 mm, respectively. Detectors 2 and 4 had surface areas of 90 mm × 15 mm and radii of 15 mm and 20 mm. The detectors were deployed in the same simulation scenario as previously, exposed to differing concentrations of ${}^{90}$Sr decay for a time period of 1 h, and the results are displayed in Table 5. When the surface area of the detector is increased threefold, the number of counts increased accordingly. As the diameter of the borehole is 50 mm, a modest increase in the detector radius to 20 mm approximately halves it sensitivity to ${}^{90}$Sr decay. This is a result of further water displacement as the volume of the detector increases. This will present an engineering challenge when developing a detector with this design as electronic components will need to fit in a compact space. Alternatively, it is feasible that the sensor could be designed as a probe remote from the main body of the detector. However, best practice typically demands that the sensor and processing electronics are situated closely together in order to avoid creating additional noise and interference in the detector.

When compared with the results in Table 1 and Table 3, it is clear that the large surface area of this design is beneficial to the detector sensitivity to ${}^{90}$Sr. In a 24-h period, it would be possible to monitor ${}^{90}$Sr to below the required level for the drinking water guideline level. However, it suffers from similar deficiencies to the planar GaAs detectors in terms of its ability to stop high energy beta decay due to the thickness of its sensitive layer, 6 μm. If the detector is assumed to have the same counts of background as previous detectors, then the limits of detection will be found as in Table 6. It is likely that the majority of beta particles which are incident on the detector travel from the immediate vicinity and the reduction in water volume each unit of surface area the detector is exposed to is not statistically significant. This performance is favourable when compared to the previous CdTe and GaAs detectors, and a flexible detector may indeed allow for rapid screening of beta emitting radionuclides in groundwater boreholes. Future work will focus on validation of the simulation results presented here. Development is under way to prototype a real-world detector which will utilise the CdTe and GaAs sensors modelled in this research. Laboratory tests replicating the experiments presented will allow for the validation of the Monte Carlo simulations with real data.

## 6. Conclusions

An in-situ ${}^{90}$Sr detector was modelled to examine its potential performance and sensitivity in a real-world scenario. This technique requires the detector to be deployed directly into a groundwater borehole, where it directly counts beta decay from radionuclides. This will eliminate the requirement for chemical separation and pre-treatment necessary for other techniques, such as liquid scintillation counting. Geant4 was used to model a groundwater borehole scenario, and the deployment of the detector into water contaminated with ${}^{90}$Sr, ${}^{90}$Y, ${}^{137}$Cs, ${}^{14}$C, and ${}^{40}$K. A 10 mm × 10 mm × 1 mm CdTe sensor was contrasted with a 20 mm × 20 mm × 3 μm GaAs sensor.

The CdTe detector was found to be sensitive to strong beta emitters, ${}^{907}$Sr, ${}^{90}$Y, ${}^{137}$Cs, and ${}^{40}$K, with that sensitivity increasing with the average emission energy of the radionuclide in question. However, there was no sensitivity to ${}^{14}$C, a weak emitter, but one which contributes significantly to the total beta counts found at Sellafield. The GaAs sensor was more sensitive to each radionuclide but was similarly blind to weak beta emitters. Therefore, such a detector is unlikely to be effective as a counter for total beta activity. This is a beneficial outcome when the detector is designed to target ${}^{90}$Sr decay. An alternative way to increase the surface of the detector is to utilise flexible sensors which can be wrapped around a hemispherical surface. A CdTe detector was modelled based on the concept of wrapping a flexible sensor around the surface of a cylinder with a radius of 15 mm and a sensitive surface surface area of 90 mm × 15 mm. This approach could be appropriate for rapid gross beta measurement in a borehole.

The limit of detector for the CdTe sensor and ${}^{90}$Sr decay was found to be 323 BqL${}^{-1}$ after a 1-h measurement and 66 BqL${}^{-1}$ after a 24-h measurement. Meanwhile, the GaAs sensor had a lower limit of detection of 91 BqL${}^{-1}$ and 18 BqL${}^{-1}$ in 1 and 24 h, respectively. Existing techniques are capable of examining ${}^{90}$Sr decay below the World Health Organisation’s drinking water guideline limit of 10 BqL${}^{-1}$. Equivalent performance in the real world would have the benefit of allowing decommissioning sites to examine the data quickly to determine whether beta activity is present or not and rapidly respond to unexpected spikes in groundwater activity resulting from new leaks. This could be used to complement existing techniques and allow for a selective approach with liquid scintillation counting used for closer examination of ${}^{90}$Sr activity.

## Figures and Tables

**Figure 1 sensors-21-00984-f001:**
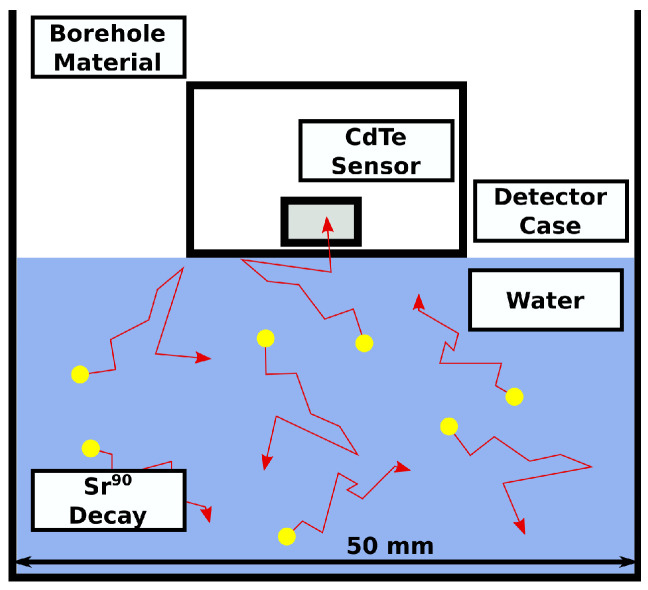
The simulation scenario illustrated. The detector is deployed to the surface of contaminated groundwater. Decaying radionuclides are randomly distributed throughout the water.

**Figure 2 sensors-21-00984-f002:**
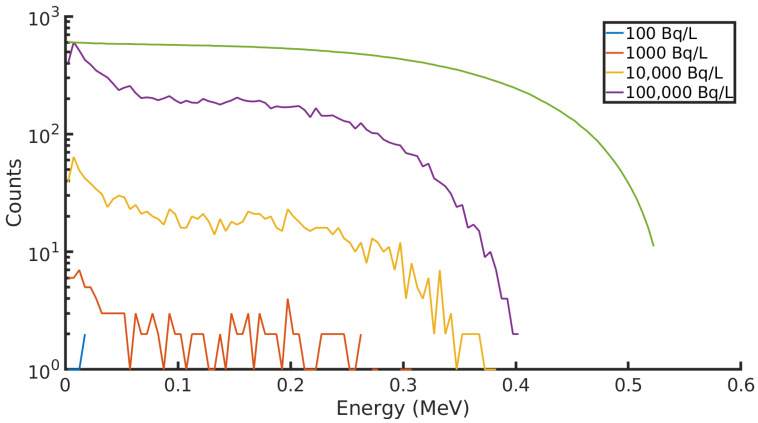
${}^{90}$Sr counted in a CdTe detector in a groundwater borehole simulation at different activities.

**Figure 3 sensors-21-00984-f003:**
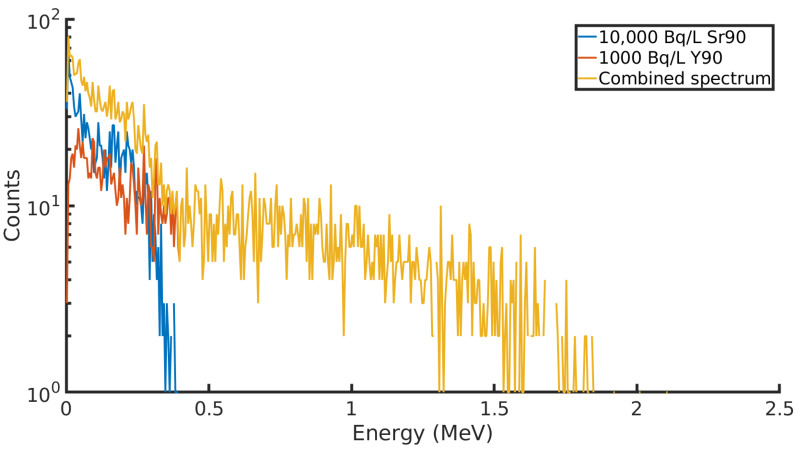
${}^{90}$Sr and ${}^{90}$Y counts recorded in a CdTe detector in a groundwater borehole simulation. The combined spectrum is plotted in yellow.

**Figure 4 sensors-21-00984-f004:**
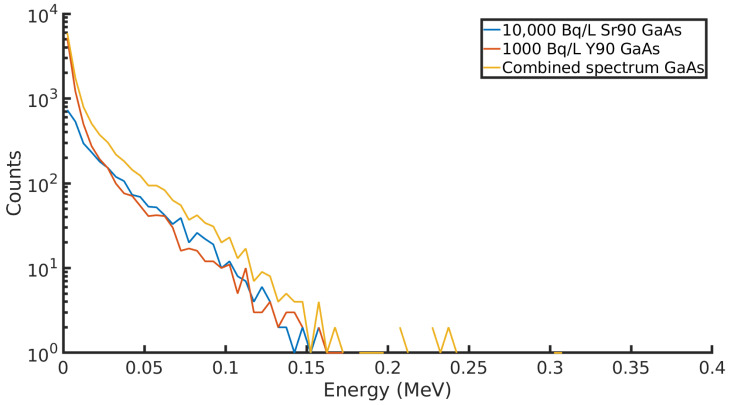
${}^{90}$Sr and ${}^{90}$Y counts recorded in a GaAs detector in a groundwater borehole simulation. The combined spectrum is plotted in yellow.

**Figure 5 sensors-21-00984-f005:**
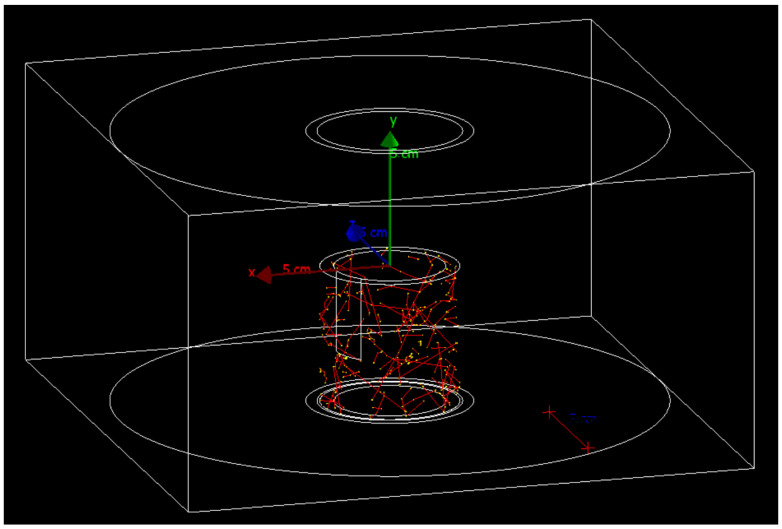
A flexible CdTe detector, white rectangle, applied to the surface of a cylindrical detector, as depicted in the Geant4 simulation with decaying ${}^{90}$Sr, yellow dots, and beta particles, red lines. Some surfaces have been hidden for clarity.

**Table 1 sensors-21-00984-t001:** The hypothetical number of counts observed in the detector for different radionuclides and activities after a 1-h counting period.

Activity BqL${}^{-1}$	${}^{90}$Sr Counts	${}^{90}$Y Counts	${}^{40}$K Counts	${}^{14}$C Counts	${}^{137}$Cs
100,000	12,752	242,593	162,060	0	78,013
10,000	1287	24,246	16,260	0	7842
1000	136	2438	1620	0	792
100	10	246	172	0	73
10	1	21	18	0	10
1	0	1	0	0	1

**Table 2 sensors-21-00984-t002:** The limit of measurement for ${}^{90}$Sr in the cadmium telluride (CdTe) detector for increasing lengths of measurement.

Length of Measurement	Limit of Detection (BqL${}^{-1}$)
10 min	799
1 h	323
24 h	66
48 h	46
168 h	25

**Table 3 sensors-21-00984-t003:** The counts observed in a gallium-arsenide (GaAs) sensor for various radionuclides found at Sellafield, and the percentage increase compared to the CdTe sensor.

Activity BqL${}^{-1}$	${}^{90}$Sr Counts	${}^{90}$Y Counts	${}^{40}$K Counts	${}^{14}$C Counts	${}^{137}$Cs Counts
100,000	28,428 (+222%)	834,089 (344%)	505,585 (311%)	0	106,997 (137%)
10,000	2859 (+222%)	83,203 (343%)	50,438 (310%)	0	10,563 (134%)
1000	321 (+236%)	8130 (333%)	5080 (313%)	0	1051 (133%)
100	32 (+320%)	802 (326%)	506 (294%)	0	106 (145%)
10	3	81	58	0	7
1	1	15	10	0	0

**Table 4 sensors-21-00984-t004:** The limit of measurement for ${}^{90}$Sr in a 10 mm × 10 mm × 3 μm gallium-arsenide (GaAs) detector (a) and a 35 mm × 35 mm × 30 μm GaAs detector (b) for increasing lengths of detection.

Length of Measurement	Limit of Detection (BqL${}^{-1}$) (a)	Limit of Detection (BqL${}^{-1}$) (b)
0.17 h	221	18
1 h	91	7
24 h	18	1
48 h	13	1
168 h	7	1

**Table 5 sensors-21-00984-t005:** The counts observed in a different flexible CdTe sensors deployed in groundwater with varying concentrations of decaying ${}^{90}$Sr. Each detector had the same material composition but different geometry.

	Detector 1	Detector 2	Detector 3	Detector 4
Dimensions (mm)	30 × 15	90 × 15	30 × 15	90 × 15
Radius (mm)	15	15	20	20
100,000 BqL${}^{-1}$	47,290	141,953	26,093	78,430
10,000 BqL${}^{-1}$	4729	14,134	2649	7932
1000 BqL${}^{-1}$	438	1373	278	782
100 BqL${}^{-1}$	41	117	27	68
10 BqL${}^{-1}$	7	17	2	11
1 BqL${}^{-1}$	0	0	0	1

**Table 6 sensors-21-00984-t006:** The limit of measurement for ${}^{90}$Sr in the flexible CdTe detector for increasing lengths of measurement.

Length of Measurement	Limit of Detection (BqL${}^{-1}$)
0.17 h	51
1 h	20
24 h	4
48 h	3
168 h	2

## Data Availability

Data of this research is available upon request via corresponding author.

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
