# Peer review of "The Simulated Characterization and Suitability of Semiconductor Detectors for Strontium 90 Assay in Groundwater"

_sensors, 2021, doi:10.3390/s21030984_

Round 1

Reviewer 1 Report

see attached file

Reviewer 2 Report

The article discusses the Characterisation and Suitability of Semiconductor Detectors for Strontium 90 Assay in Groundwater.

The introduction part can be improved by providing more relevant references and this part must be more interesting to the audience.

The authors correlating each analysis to support the experimental results but need to improve with more evidence with reference and overall English writing can be improved. The whole manuscript can be organized with more relevant information with reasonable evidence. However, the following comments must be addressed prior to the publication.

Comments:

  1. Title: Characterisation must be replaced with Characterization – Typo Error
  2. Keywords: Authors should be very specific on the keywords, should replace the word - strontium; beta; radiation; detection with more specific keywords from the present research in the paper (Ex: strontium-90, Geant4 simulation).
  3. Ln No: 48 : results in a more time294 commentsly and cost effective manner – Check the sentence and modify
  4. Ln No: 50 - Acrorad (Japan) – must include a reference
  5. Ln No 68: The sentence must be rewritten “This, and the continuous nature of beta emission, has presented difficulties when it comes to designing beta detectors”.
  6. Ln No 94: FTFP BERT – Must write the abbreviation
  7. Ln No: 158 – WHO -? Abbreviation?
  8. Ln No 194 : The previous sections assesses a 10 x 10 x 1 mm CdTe sensor, but other sensors types are available. One such sensor is a gallium-arsenide------------------ The sentence can be re written------   

Reviewer 3 Report

In the manuscript the Authors examine the potential deployment of a CdTe detector for 90Sr measurement in groundwater boreholes at nuclear decommissioning sites, with special attention to the Sellafield case. Results from GEANT4 simulations are presented for different arrangements of detectors based on CdTe material, as well as on the other semiconductor GaAs. All of these devices have potential beneficial impact and limitations as tools for radioactive environmental monitoring and are worth to be investigated further. According to the results of this study the detection limit of 10 Bq/L, recommended by the World Health Organization for drinking water, is hard to be achieved with such devices, but still possible, especially for larger area detectors. However when this goal is achieved the radiation from 90Sr beta decay cannot be distinguished from that generated by the 90Yb decay.
Despite this research is justified by a solid case, there are important weak points in the manuscript that need to be addressed. A make a list of the main issues below.
1. The GEANT4 simulations are not validated by any experimental datum. In this way no evaluation of the uncertainties in the final extracted sensitivities can be given, which is in fact the case. How can we trust on the results? The Authors should compare the results of specific simulations with a real CdTe detector with the same size (10 mm x 10 mm x 1 mm), or at least the GaAs (20 mm x 20 mm x 0.006 mm) to get confidence on their simulations. In this way they could also propose different geometries and materials with a certain degree of confidence.
2. The Authors seem not to take this drawback into account, since no comment is found in the manuscript and the conclusions do not consider the validation of the simulations as a perspective of the future activity.
3. Despite the manuscript is proposed for a review article, I do not see the content going in such direction. Instead, it presents a very specific and incomplete (for the reasons expressed on item 1) simulation work.
4. It is not clear to me why one should use detection systems based on a single detector for making groundwater boreholes monitoring. For example an array of 10 independent detectors will count ten times faster and get better sensitivity. The Authors should comment this “poor man” simple argument.
5. It is not clear why, the comparison between CdTe and GaAs is only performed for quite different geometries, namely (10 mm x 10 mm x 1 mm) for the CdTe and (20 mm x 20 mm x 0.006 mm) for the GaAs. Is it not possible to ask the producer to build detectors with different sizes, tailored to the need of this monitoring activity?
6. Nothing is said about the sources of noise and the amount of background assumed for the real detectors, so it is not clear what kind of improvements could be envisaged.
7. The manuscript is full of typos and some sentences are maybe lacking words. In general not much care is paid to the style and even to the language, which is a strange feature as the Authors are native English speakers. My impression is that they did not revise the manuscript very carefully.
I would have tens of little comments and typos, but before going through all of them, I would like to express my general view. I consider the validation of the simulation indispensable for a high standard publication. Without a convincing answer to the above listed items I do not recommend the Editor for a publication of this manuscript in Sensors. I would anyway be happy to re-evaluate un upgraded version of it, which takes these points into account.

Round 2
